# NanoDefiner e-Tool: An Implemented Decision Support Framework for Nanomaterial Identification

**DOI:** 10.3390/ma12193247

**Published:** 2019-10-04

**Authors:** Raphael Brüngel, Johannes Rückert, Wendel Wohlleben, Frank Babick, Antoine Ghanem, Claire Gaillard, Agnieszka Mech, Hubert Rauscher, Vasile-Dan Hodoroaba, Stefan Weigel, Christoph M. Friedrich

**Affiliations:** 1Department of Computer Science, University of Applied Sciences and Arts Dortmund (FH Dortmund), 44227 Dortmund, Germany; raphael.bruengel@fh-dortmund.de (R.B.); johannes.rueckert@fh-dortmund.de (J.R.); 2Material Physics Research, BASF SE, 67056 Ludwigshafen, Germany; wendel.wohlleben@basf.com; 3Institute of Process Engineering and Environmental Technology, Technische Universität Dresden (TU Dresden), 01062 Dresden, Germany; frank.babick@tu-dresden.de; 4R&I Centre Brussels, Solvay S.A., 1120 Brussels, Belgium; antoine.ghanem@solvay.com; 5DG JRC, European Commission, 21027 Ispra, Italy; claire.gaillard@ec.europa.eu (C.G.); agnieszka.mech@ec.europa.eu (A.M.); hubert.rauscher@ec.europa.eu (H.R.); 6Division 6.1 Surface Analysis and Interfacial Chemistry, Bundesanstalt für Materialforschung und -prüfung (BAM), 12205 Berlin, Germany; dan.hodoroaba@bam.de; 7Institute of Food Safety, RIKILT Wageningen UR, 6708 WB Wageningen, The Netherlands; stefan.weigel@wur.nl; 8Institute for Medical Informatics, Biometry and Epidemiology (IMIBE), University Hospital Essen, 45122 Essen, Germany

**Keywords:** nanotechnology, nanomaterial, particulate material, particle measurement, decision support, expert system, EC nanomaterial definition

## Abstract

The European Commission’s recommendation on the definition of nanomaterial (2011/696/EU) established an applicable standard for material categorization. However, manufacturers face regulatory challenges during registration of their products. Reliable categorization is difficult and requires considerable expertise in existing measurement techniques (MTs). Additionally, organizational complexity is increased as different authorities’ registration processes require distinct reporting. The NanoDefine project tackled these obstacles by providing the NanoDefiner e-tool: A decision support expert system for nanomaterial identification in a regulatory context. It provides MT recommendations for categorization of specific materials using a tiered approach (screening/confirmatory), and was constructed with experts from academia and industry to be extensible, interoperable, and adaptable for forthcoming revisions of the nanomaterial definition. An implemented MT-driven material categorization scheme allows detailed description. Its guided workflow is suitable for a variety of user groups. Direct feedback and explanation enable transparent decisions. Expert knowledge is held in a knowledge base for representation of MT performance criteria and physicochemical particle type properties. Continuous revision ensured data quality and validity. Recommendations were validated by independent case studies on industry-relevant particulate materials. Besides supporting material identification and registration, the free and open-source e-tool may serve as template for other expert systems within the nanoscience domain.

## 1. Introduction

The European Commission’s recommendation on the definition of nanomaterial (2011/696/EU) (EC NM recommendation) [1] relies on the knowledge of the number-based particle size distribution of a material. Depending solely on the median value of the number-based particle size distribution, below and equal to 100 nm, or above 100 nm, the materials will be categorized into nano- or non-nanomaterials. The European project NanoDefine (https://www.nanodefine.eu/ (access 2019-08-16)) has been exclusively dedicated to the evaluation of particle size of representative materials with the aim of categorization of a material as nano or non-nano. For this, a newly developed tiered approach of suitable characterization methods (henceforth referred to as *methods*) has been proposed. In this context, these are defined either as screening or as confirmatory methods. Beside the measurement technique (MT), a method also comprises suited sample preparation and data analysis procedures. The validation of methods has been established by using well-defined quality control materials (QCMs) and representative test materials (RTMs) [2]. Systematic results on particle size measurement, obtained with practically all methods able to determine particle size (see later details), were cross-checked against values obtained by electron microscopy which was considered to be the reference method. The specific purpose of a method as used in the NanoDefine tiered approach consists of the determination of the median value of the number-based size distribution ($x_{50,0}$) of constituent particles of a particular material, after successful, homogeneous dispersion. The number-based particle size distribution is a list of values, defining the relative amount by numbers of particles according to their size, following ISO/TS 19590:2019 [3].

A major project outcome is the NanoDefiner: A decision support framework, capable of assisting expert as well as non-expert users in the categorization and registration process of potential nanomaterials for regulatory purposes. Specific materials are described using a Material Categorization Scheme (MCS) [4,5]. MTs suitable for particle size analysis of a specific material are recommended, accompanied by detailed and tailored explanation. Recommendations largely rely on a material characterization scheme, conducted with different MTs on various QCMs and RTMs selected within the NanoDefine project [6,7].

Investigating capabilities of MTs, suitable for nanomaterial identification, is necessary for material research and regulation [8,9,10]. However, thus far no single MT is suitable to identify all kinds of potential nanomaterials [8]. Selection of suitable MTs for identification of a potential nanomaterial usually requires expert knowledge which is not always available. Moreover, for this purpose currently no other decision support frameworks or recommendation systems are available. This manuscript gives a comprehensive introduction to the *NanoDefiner e-tool* as an implementation of the NanoDefiner framework, following the concepts and terms [11] used in the EC NM recommendation.

The e-tool (https://fh.do/nanodefiner/ (access 2019-08-16)) [12] was implemented as an expert system for material categorization and aims to support material registration processes. Therefore, it provides MT recommendations for the assessment of specific materials. Its material categorization approach is based on the NanoDefine Decision Support Flow Scheme (NanoDefine DSFS) [13]. Technical details on its extensible knowledge base and rule-based multi-criteria decision making for recommendation are outlined. Interoperability is illustrated, referring to supported analysis data formats for automated categorization of materials as nano or non-nano. As free and open-source software its customizability, extensibility, and reusability for diverse purposes are highlighted. To conclude, a brief summary of the e-tool is given and download resources are listed. Independent industrial case studies [14] are referenced, confirming its consistency with the NanoDefine DSFS. Finally, its potential for manufacturers of materials and MTs as well as for independent analytical laboratories is pointed out.

## 2. Materials and Methods

One important characteristic of the NanoDefiner framework is that it has been developed based on a comprehensive set of representative materials as well as taking into account practically all currently available MTs capable of evaluating particle size distribution. Its implementation, the NanoDefiner e-tool, relies on expert knowledge represented in the NanoDefiner framework which has been cultivated throughout the NanoDefine project.

In the following, details on the materials and MTs tested are presented. In addition, approaches for knowledge engineering and knowledge management are outlined.

### 2.1. Materials

First, a set of six materials has been employed as QCMs [15] (see Table 1), i.e., nanoparticles which are well-defined and therefore well-suited to qualify the analytical figures of merit of various sizing techniques. Typical colloidal gold, silica, polystyrene, and silver nanoparticles have been selected as representative for the monodisperse, spherical model nanoparticles as often reported in the literature. Furthermore, trimodal, spherical silica and polystyrene nanoparticle samples have been considered. While for the first type of nanoparticles it is expected that all sizing MTs perform accurately enough, the multimodal nanoparticles were destined as advanced test for the qualified MTs, so that the limitations of some MTs become quickly noticeable.

A set of 11 RTMs [15] (see Table 1) has been carefully selected as proxies for relevant classes of “real-world” nanoparticulate materials which pose the highest challenges on all the MTs. Among these challenges, following material properties can be enumerated: High polydispersity, non-spherical constituent particles, aggregated state, fractal-like aggregates, low dispersibility/dispersing efficiency (i.e., still aggregated state of somewhat reduced size even with a lot of input energy). Furthermore, the impact of measurement conditions on granulometric state (e.g., by dissolving after extreme high dilution) or impact of material properties on measurement sensitivity (e.g., low detection sensitivity for Silicon in single particle Inductively Coupled Plasma Mass Spectrometry (spICP-MS) or for Silica in Particle Tracking Analysis (PTA)) should be also considered. Broad particle size polydispersities, complex shapes and high degree of agglomeration are characteristics of the selected RTMs.

The systematic consideration of these challenging particulate materials is one of the most valuable outcomes of the NanoDefine project as well as the NanoDefiner framework and differentiates this project from the large majority of the published studies on nanoparticle size measurement, which are often applied on “user-friendly” spherical, monodisperse particles. Details on NanoDefine materials are given in previous publications [6]. Properties needed for exact description of materials within the NanoDefiner framework are covered by the MCS [4,5] of the NanoDefiner e-tool and itemized in its knowledge base (see Appendix A).

### 2.2. Methods

As far as the MTs included in the NanoDefiner framework (see Table 2) are regarded, practically all nanoparticle sizing techniques were considered after systematical evaluation in the NanoDefine project (see Figure 1):
(i)*Counting* techniques, which are in favor to the application of the EC NM recommendation, i.e., Electron Microscopy (EM) (Scanning Electron Microscopy (SEM), Transmission Electron Microscopy (TEM), including also derivatives such as mini TEM (miniTEM)), Atomic Force Microscopy (AFM), PTA, spICP-MS, and Tunable Resistive Pulse Sensing (TRPS),(ii)*Fractionation* techniques, i.e., Field Flow Fractionation (FFF), centrifugation techniques centrifugation techniques (more versions: Disc centrifuges with turbidity detector, cuvette centrifuges with turbidity detector, cuvette centrifuges with refractive index measurement) and Differential Electrical Mobility Analysis (DEMA),(iii)*Spectroscopic (Ensemble)* techniques (Dynamic Light Scattering (DLS), Small-Angle X-ray Scattering (SAXS), Ultrasonic Spectroscopy (USSP), Angular Light Scattering (ALS)), and(iv)*Integral* techniques such as X-ray Diffraction (XRD) and Brunauer–Emmett–Teller (BET) for determination of Volume-Specific Surface Area (VSSA) (with knowledge of skeleton density [16]: The ratio between sample mass and volume, including closed pore volume, and excluding open pore volume and void spaced between particles within the bulk space) have been thoroughly characterized with respect to their analytical performance criteria.

As a result, a considerable set of performance criteria [17,18] has been identified and quantified as characterizing comprehensively all MTs considered [19]. These criteria are itemized in the knowledge base of the NanoDefiner e-tool (see Appendix A).

One fundamental aspect in the NanoDefiner framework implementation strategy is the tiered approach concept (see Table 2) based on selective application of MTs of increasing complexity and complementary measurement principles. MTs as part of Tier 1 are screening techniques being cost-efficient, widely available as well as robust enough. For more complex particles, mostly sophisticated MTs such as EM are unavoidable to be considered to be confirmatory Tier 2 MTs.

An added value offered to the user by the NanoDefiner framework consists of accompanying guidance on sampling, sample preparation, data evaluation, plausibility checks and minimum performance requirements by providing adequate documented procedures.

Again, it should be noticed that in the following the term “method” considers a characterization method, meaning an MT together with prior sample preparation as well as analysis of measurement data. For each material, dedicated sample preparation protocols have been developed in the NanoDefine project to ensure for all laboratories a uniform state of dispersion [20,21].

### 2.3. Decision Support Flow Scheme

Insight obtained from the different NanoDefine project work packages are pooled in the NanoDefine DSFS [13] that aims to provide the fastest way towards a reliable material categorization, preferably avoiding cost-intensive and time-consuming analyses. Besides being economically viable, its pragmatic approach also fulfills regulatory obligations by yielding certain decisions. A simplified representation of the NanoDefine DFSF is depicted in Figure 2.

Several decision nodes guide the user towards a categorization decision on a specific material. A first basic decision addresses materials groups which are *per se* to be categorized as nano or non-nano by the EC NM recommendation. As many materials cannot be ascribed directly to such groups, the user may either directly choose to apply a confirmatory Tier 2 method such as for instance EM, or refer to screening Tier 1 methods. For the latter, the NanoDefine DSFS provides a powder route as well as a dispersion route, also described in detail in [13,22,23]. Either may be traversed depending on the trade form and dispersibility of the material to be analyzed. However, both Tier 1 routes may result in the intermediate identification of a “borderline material”.

In particular, for borderline cases for particles with an $x_{50,0}$ around 100 nm, Tier 2 MTs are needed. The nano/non-nano cutoff value of the $x_{50,0}$ (as resulted from investigations in the NanoDefine project) is set at 250 nm to escalate from Tier 1 to Tier 2. This value can be adapted at a later stage within the e-tool as a possible consequence of upcoming, improved systematic studies. Alternatively, another Tier 1 method using a different MT may be performed to cross-check the result (henceforth referred to as *plausibility check*).

It should be noted that several researcher groups worldwide are working on different schemes [24,25,26] for the characterization/categorization of nanomaterials.

### 2.4. Knowledge Engineering

Knowledge engineering refers to all aspects having an impact on the process of designing, implementing, and maintaining a knowledge-based system. Beside technical and scientific aspects, especially social aspects have a substantial impact. Expert systems, of which MYCIN [27] is a well-known example, are a differentiation of knowledge-based systems [28] that simulate inference of domain experts. To do so, such systems incorporate respective expert knowledge.

In the widest sense of the word, expert knowledge refers to any kind of knowledge an expert activates to fulfill a specific task, e.g., deciding which MT might be suitable for the analysis of a specific material. This comprises explicit knowledge, for instance acquired from literature such as Standard Operating Procedures (SOPs). Though, also tacit knowledge [29] manifested in intuitive and creative behavior plays a role in inferring processes. A common way to make knowledge available for an expert system is to cumulate and persist it in a knowledge base [30]. There, it is stored in a structured and formalized, and thus machine-processible manner. In rule-based expert systems, methodical expert knowledge is also represented in form of production rules [31] that are executed to infer decisions, involving expert knowledge stored in a knowledge base.

As the NanoDefiner e-tool was conceived to be a rule-based expert system, hence for conception, development, and maintenance methods of knowledge engineering were applied. This usually involved groups of five to ten nanoscience domain experts attending workshops and virtual meetings with two to three computer scientists functioning as knowledge engineers. The conducted knowledge engineering process (see Figure 3) adapted a classical approach [32], comprising the phases of 1. *Identification*; 2. *Conceptualization*; 3. *Formalization*; 4. *Implementation*; 5. *Testing*; and 6. *Revision*:
*Identification*: Experts from academia and industry discussed important aspects, goals, problems, and resources with knowledge engineers. This phase yielded requirements for the e-tool and drafted an idea on how to implement them. Beside the experts themselves, the main sources of knowledge were identified as: (i) the NanoDefiner DSFS [13]; (ii) filled out MT performance criteria tables [18] with findings relying of literature analysis and laboratory experiments on NanoDefine QTMs and RTMs [6,7]; and (iii) the MCS for description of physicochemical properties of particle types [4,5].*Conceptualization*: Explicit concepts on the e-tool regarding its workflow, decision making on MT recommendations for the analysis of specific materials, and categorization of potential nanomaterials were created by knowledge engineers. This involved consideration of the main knowledge sources and requirements identified before.*Formalization*: Concepts were formalized by knowledge engineers to create a knowledge base and production rules [33] in both of which conceptual expert knowledge is represented. Explicit knowledge was transformed into a structured and formalized form and persisted in the knowledge base to be processible programmatically.*Implementation*: Based on the formalized knowledge an expert system implementation of the e-tool was developed by knowledge engineers to traverse the workflow and execute production rules, involving knowledge of MT performance in the knowledge base and knowledge of a specific materials entered by users via an implementation of the MCS.*Testing*: Experts and knowledge engineers performed tests on the e-tool implementation to verify its behavior and to identify bugs, problems, incorrectness, inconsistency, and ambiguity regarding the underlying NanoDefine DSFS, MT recommendations, and categorization.*Revision*: Based on test findings and estimation of experts, a revision was performed by knowledge engineers. Results of the phases *Identification*, *Conceptualization*, *Formalization*, and *Implementation* were adjusted and extended, resulting in a new *Testing* phase.

### 2.5. Knowledge Management

Knowledge management is a broader term for multidisciplinary approaches to generate, distribute, use, and revise knowledge. When constructing an expert system, a key challenge during the knowledge engineering process is to realize and gather tacit expert knowledge, which experts may initially not be able to articulate and would refer to as intuition. During the development of the NanoDefiner e-tool, alternation of *Testing* and *Revision* phases (see Figure 3) often resulted in revelation of tacit knowledge when experts were facing unexpected behavior. Knowledge management conducted during these cycles (see Figure 4) was oriented towards the 1. *Socialization*, 2. *Externalization*, 3. *Combination*, and 4. *Internalization* (SECI) model [29,34]:
*Socialization* was performed during tests, single or multiple experts tested the e-tool and gained experience on how it worked.*Externalization* was conducted during revision meetings, the group of experts were interviewed, shared their experiences, discussed unexpected behavior, and concluded required behavior as a result of tacit knowledge being transformed to explicit knowledge.*Combination* then took place by achieving a consensus on required behavior of a revision, now relying on explicit knowledge.*Internalization* was applied revising the e-tool, incorporating prior tacit knowledge as newly gained explicit knowledge. The resulting e-tool version built the base for a new SECI cycle.

This cycle was performed multiple times and lead to constant gain and refinement of explicit knowledge as well as to optimized behavior of the e-tool implementation. An example of refinement may be the evolution of the borderline range for a material analyzed via Tier 1 methods [22,23].

## 3. Results

The NanoDefiner e-tool accommodates analysts with different levels of expertise, providing a guided workflow. A tiered approach divides MTs into two tiers: Tier 1 comprising screening MTs, and Tier 2 comprising confirmatory MTs. Information regarding the categorization process of materials with one or multiple types of particles is aggregated in a single dossier for a distinct purpose (e.g., for a specific regulation such as Registration, Evaluation, Authorization and Restriction of Chemicals (REACH) [35]).

Physicochemical properties of particulate materials, relevant for an adequate characterization [10] and regulatory aspects [36,37,38], are described via an MCS [4,5]. Particulate material descriptions may be derived from NanoDefine materials or custom templates. Expert knowledge of MT performance as well as physicochemical properties of QCMs and RCMs is structured, formalized, and persisted in an extensible and robust knowledge base. Rule-based decision making uses this knowledge to infer recommendations on MTs suitable for the analysis of specific materials described by the user. Live feedback and explanation on MT recommendations establish comprehensibility and transparency. Supported analysis data formats can be imported for an automated categorization. Support of custom analysis data formats can be extended using an Application Programming Interface (API).

Detailed descriptions on respective results are outlined further in the following.

### 3.1. Workflow

The simplified underlying guided workflow for material categorization (see Figure 5) comprises seven stages: 1. *Dossier creation* for a material sample; 2. *Particle description* of the particle type(s) in the sample; 3. *Measurement technique recommendation and method selection*; 4. *Tier 1/Tier 2 method application* of a respective method in the laboratory; 5. *Analysis data import* of yielded analysis data; 6. *Material categorization as nano/non-nano*; and 7. *Report generation* for further use. Following, a workflow walkthrough for a material sample with one type of particles is described that can be tracked by examining the enclosed screencast (see Appendix A):
*Dossier creation*: The first step in the material categorization workflow is the creation of a dossier, which is the highest-level entity in the e-tool and comprises a material sample (consisting of one or more particle types), applied methods, and a report. In addition to choosing a name for the dossier, users state a purpose (e.g., fulfilling obligations from a specific regulation [39]) as well as whether the sample consists of one (*mono-type* sample) or multiple (*multi-type* sample) types of particles, both of which determining the availability of MTs in later stages of the workflow. The following steps refer to the *mono-type* case.*Particle description*: After creating the dossier, the user describes the physicochemical particle type properties of the *mono-type* material sample via the MCS, a multi-page form allowing specification of particle properties. The live feedback on MT suitability and recommendations is one of the core features of the NanoDefiner e-tool, backed by rule-based decision making based on the information of the knowledge base.*Measurement technique recommendation and method selection*: Methods are performed to document the process of analyzing the described sample with one of the available MTs. In this step of the workflow, users can make use of Tier 1 (screening) or Tier 2 (confirmatory) methods based on a list of configured MTs and their suitability for the material sample, as well as choosing the pre-processing protocol and name for the method.*Method application*: The next step is performed outside of the NanoDefiner e-tool. The user applies one or more selected methods. Methods may also be applied multiple times.*Analysis data import*: When analysis results are available the user can upload analysis data or manually state results. Optionally, a percentaged MT uncertainty (not be confused with measurement uncertainty, associated with the result obtained with an MT) can be stated. For supported analysis data formats direct categorization and visualization is available.*Material categorization*: After uploading analysis results, a nano, non-nano, or intermediate borderline decision (see Figure 2) is made for the material. For Tier 1 methods the user will be informed in case a Tier 2 confirmation or plausibility check is necessary (see Figure 5). For supported formats, plots will be generated and displayed according to ISO 9276-1:1998 [40].*Report generation*: Once at least one method has been completed (i.e., analysis results have been uploaded), a Portable Document Format (PDF) report can be generated for the dossier. A report contains detailed information on the dossier, its sample, and a custom set of applied methods, including analysis and categorization results. Additionally, Appendix A on MT availability and suitability with explanation is provided. Plain reports are PDF/A-1-compliant following ISO 19005-1:2005 [41], allowing long-term preservation. Further data supporting evidence (e.g., raw analysis data) may be attached.

During the whole workflow, the integrated NanoDefine Methods Manual [42,43,44,45,46] can be accessed for consultation in case that further information on concepts or requested input is needed. Additionally, for direct access of relevant manual sections the e-tool embeds references, e.g., alongside of respective form elements of the MCS.

### 3.2. Knowledge Base

The spreadsheet-based knowledge base is documented and maintainable by non-computer scientists. Continuous evaluation and multiple revision cycles ensured validity of stored knowledge. MTs were assessed by experts from industry and academia via literature analysis as well as by systematic measurements and analysis on materials [6]. Currently, it comprises 17 QCM and RTM property profiles (see Table 1), and 16 MT default performance criteria profiles (see Table 2).

For description of MTs, well-defined attributes were derived from filled out templates for quantitative and qualitative description of MT performance criteria [17,18], as well as from the MCS for description of particulate material properties [4,5]. Particles are described via 21 attributes on physicochemical features according to the MCS (e.g., trade form, dispersibility, stable temperature range) (see Appendix A), MTs are described via 83 attributes on material property support, measurement performance, and technical/economic aspects (e.g., particle shape, working size range, cost efficiency) (see Appendix A).

The knowledge base setup consists of dictionaries and associated sheets that form logical pairs. A dictionary defines a set of attributes for which value assignments are made in the related sheet. A set of value assignments represents a profile, multiple of such profiles can be described. An attribute has a unique name, a data type, and a value scope. To describe properties pre-defined data types exist for representation of character strings (*string*), value sets (*set*), decimals (*decimal*), decimal intervals (*interval*), Boolean values (*binary*), and value scales (*scale*).

In the current version of the knowledge base the setup (see Figure 6) consists of six different data dictionaries and sheets that coexist with certain relations: (i) the *Measurement technique dictionary* and *sheet*; (ii) the *Measurement technique performance dictionary* and *sheet*; (iii) the *Material group dictionary* and *sheet*; (iv) the *Material property dictionary* and *sheet*; (v) the *Priority dictionary* and *sheet*; as well as (vi) the *Explanation dictionary* and *sheet*:
(i)*Measurement technique dictionary* and *sheet*: Administrative component for definition of MTs; contains controlling attributes.(ii)*Measurement technique performance dictionary* and *sheet*: Core component for description of default or material group-dependent MT performance criteria profiles; contains controlling and matchable attributes.(iii)*Material group dictionary* and *sheet*: Administrative component for definition of material groups for which an MT may show non-default performance; contains controlling attributes.(iv)*Material property dictionary* and *sheet*: Core component for description of particulate material property profiles; contains controlling and matchable attributes.(v)*Priority dictionary* and *sheet*: Administrative component for individual weighting of MT performance criteria attributes for decision making; contains controlling attributes only.(vi)*Explanation dictionary* and *sheet*: Administrative component for description of explanatory text fragments for explanation during decision making process; contains controlling attributes only.

A special feature of the knowledge base lies in its capability of handling missing knowledge and case-specific irrelevances in the description of MT performance criteria and material property profiles. This allows expression of lack of knowledge regarding characteristics of configured MT performance criteria and material properties described via the MCS. Also, it allows the statement that certain attributes of configured MTs are not relevant for the decision making process.

### 3.3. Rule-Based Decision Making

Recommendation of suitable MTs for a specific material via rule-based decision making is a key feature of the NanoDefiner e-tool. Particle properties stated in the MCS are processed by rules, matching them against MT property profiles described in the knowledge base. Changes in the MCS instantly trigger decision making via the Drools (https://www.drools.org/ (access 2019-08-16)) rule-processing engine that uses the novel PHREAK algorithm [47,48], built up on the Rete algorithm [49]. Inferred results are shown instantaneously (see Figure 7). Hereby, influences of changed properties are comprehensible and made transparent.

The decision making itself comprises expert knowledge, present as a combination of production rules [31] which are a common representation form for simulating cognitive behavior of experts [33]. It is robust against missing knowledge (henceforth referred to as *uncertainty* in the context of decision making). Another feature is to exclude the influence of specific particle properties that are irrelevant for the recommendation of certain MTs.

The following formal description of the decision making is a bottom-up approach and elaborates the inference process for MT recommendation for a *mono-type* sample. The special case of a *multi-type* sample is approached subsequently. Non-default MT performance is not considered in favor of reduced complexity. It starts with a description of attributes as lowest units of the decision model, brings them into a relation, illustrates how decisions are determined, and finally explains the process towards the final set of recommended MTs for a specific material:

In general, all data processed by the rule-processing engine is inserted in form of attributes. A single attribute *a* has data fields (henceforth referred to as $a_{[<\mathrm{field}\;\mathrm{name}>]}$) that incorporate a unique name $a_{\left[\mathrm{name}\right]}$, a data type $a_{\left[\mathrm{type}\right]}$, a potentially limited value scope $a_{\left[\mathrm{scope}\right]}$, and a value represented by $a_{\left[\mathrm{value}\right]}$. The value $a_{\left[\mathrm{value}\right]}$ always needs to represent data of the data type $a_{\left[\mathrm{type}\right]}$ and needs to be within the scope $a_{\left[\mathrm{scope}\right]}$.

A specific MT t∈T and a specific type of particles *p* are described each by an independent set of attributes Ω. These descriptive attributes contain matchable attributes A∈Ω (see Appendix A) and controlling attributes C∈Ω. The set of matchable attributes *A* does not intersect the set of controlling attributes *C* used exclusively for logical management on a programmatic level. Thus, for a single MT *t* there is a set of *k* matchable attributes
(1)$$A^{t}=\left\{a_{1}^{t},\ldots ,a_{k}^{t}\right\}=\Omega^{t}\backslash C^{t}$$
defined in the *Measurement technique performance dictionary* and assigned with values in its derived *sheet* of the knowledge base. For a particle type *p* there is a set of *l* matchable attributes
(2)$$A^{p}=\left\{a_{1}^{p},\ldots ,a_{l}^{p}\right\}=\Omega^{p}\backslash C^{p}$$
defined in the *Material property dictionary* of the knowledge base that is supplied with values by the user input in the MCS. Both attribute sets $A^{t},A^{p}$ are used for the inference process.

Matchable attributes $A^{p}$ of a particle type *p* are derived from a subset of matchable attributes of $A^{t}$, resulting in $|A^{p}\left|\leq \right|A^{t}|$. This is due to the circumstance that initially more MT performance criteria were documented than physicochemical particle properties retrieved from the MCS. Only attributes with identical names, types, and scopes are supposed to be matched against each other. This ensures consistent value matching of single attributes. For this purpose, the bijective relation
(3)$$ℜ=\left\{\left(a^{t},a^{p}\right)\in A^{t}\times A^{p}\;|\;a_{\left[\mathrm{name}\right]}^{t}=a_{\left[\mathrm{name}\right]}^{p},\;a_{\left[\mathrm{type}\right]}^{t}=a_{\left[\mathrm{type}\right]}^{p},\;a_{\left[\mathrm{scope}\right]}^{t}=a_{\left[\mathrm{scope}\right]}^{p}\right\}$$
is used to describe tuples of matchable attributes of $A^{t},A^{p}$. Additionally, it is ensured that for every attribute $a^{p}\in A^{p}$ also exactly one matchable attribute $a^{t}\in A^{t}$ exists
(4)$$\forall a^{p}\in A^{p}\;\exists !a^{t}\in A^{t}:\;\left(a^{t},a^{p}\right)\in ℜ$$
and hence every $a^{p}\in A^{p}$ is present in the distinct tuples $\left(a^{t},a^{p}\right)\in ℜ$, resulting in $|A^{p}\left|=|ℜ\right|$. This is due to the demand that every single particle property stated in the MCS must also be matched against a related MT property and not vice versa.

Summarizing the aforementioned construct, it is ensured that a matching of a tuple $\left(a^{t},a^{p}\right)\in ℜ$ can be conducted. However, as distinct data types exist, the matching of attributes joint in a tuple must be conducted in dependence on the underlying data type. This is established by matching methods for the data types set, interval, binary, string, decimal, and scale. Certainly, single decision points that operate on the same data type of attributes may not always demand the use of the same functions for matching. Also, single decision points may join various functions to establish the matching. This depends on the individual context of the attributes. Thus, the check of a match
(5)$$a^{t}⊢a^{p}\;\hat{=}\;a^{t}\;\mathrm{satisfies}\;a^{p}$$
in which the MT property $a^{t}$ can operate with the given particle type property $a^{p}$, will be illustrated in an abstract way. For this purpose, the abstract and data type-dependent Boolean matching functions
(6)$$\begin{array}{cc} m_{\mathrm{set}}\left((a^{t},a^{p})\right) & =\left\{\mathrm{true}\;|\;{\left(a^{t},a^{p}\right)}_{\left[\mathrm{type}\right]}=\mathrm{set},\;a^{t}⊢a^{p}\right\} \end{array}$$
(7)$$\begin{array}{cc} m_{\mathrm{interval}}\left((a^{t},a^{p})\right) & =\left\{\mathrm{true}\;|\;{\left(a^{t},a^{p}\right)}_{\left[\mathrm{type}\right]}=\mathrm{interval},\;a^{t}⊢a^{p}\right\} \end{array}$$
(8)$$\begin{array}{cc} m_{\mathrm{binary}}\left((a^{t},a^{p})\right) & =\left\{\mathrm{true}\;|\;{\left(a^{t},a^{p}\right)}_{\left[\mathrm{type}\right]}=\mathrm{binary},\;a^{t}⊢a^{p}\right\} \end{array}$$
(9)$$\begin{array}{cc} m_{\mathrm{string}}\left((a^{t},a^{p})\right) & =\left\{\mathrm{true}\;|\;{\left(a^{t},a^{p}\right)}_{\left[\mathrm{type}\right]}=\mathrm{string},\;a^{t}⊢a^{p}\right\} \end{array}$$
(10)$$\begin{array}{cc} m_{\mathrm{decimal}}\left((a^{t},a^{p})\right) & =\left\{\mathrm{true}\;|\;{\left(a^{t},a^{p}\right)}_{\left[\mathrm{type}\right]}=\mathrm{decimal},\;a^{t}⊢a^{p}\right\} \end{array}$$
(11)$$\begin{array}{cc} m_{\mathrm{scale}}\left((a^{t},a^{p})\right) & =\left\{\mathrm{true}\;|\;{\left(a^{t},a^{p}\right)}_{\left[\mathrm{type}\right]}=\mathrm{scale},\;a^{t}⊢a^{p}\right\} \end{array}$$
are introduced. These do not represent the exact condition for a match $a^{t}⊢a^{p}$ but a matching operation in case of a certain data type. For instance, for a certain single decision point the abstract matching function $m_{\mathrm{set}}\left(\;\cdot \;\right)$ may represent a superset check $a^{t}⊇a^{p}$ that will check whether the MT property $a^{t}$ is able to operate on all elements of the given particle property $a^{p}$. For another single decision point, it may represent an intersection check $a^{t}\;\cap \;a^{p}$ that will check whether the MT property $a^{t}$ is able to operate on at least one elements of the given particle property $a^{p}$. For further explanation of the decision making, the data type-dependent matching functions will be encapsulated in the Boolean major matching function
(12)$$m\left((a^{t},a^{p})\right)=\left\{\begin{array}{cc} m_{\mathrm{set}}\left((a^{t},a^{p})\right), & {\left(a^{t},a^{p}\right)}_{\left[\mathrm{type}\right]}=\mathrm{set}\\ m_{\mathrm{interval}}\left((a^{t},a^{p})\right), & {\left(a^{t},a^{p}\right)}_{\left[\mathrm{type}\right]}=\mathrm{interval}\\ m_{\mathrm{binary}}\left((a^{t},a^{p})\right), & {\left(a^{t},a^{p}\right)}_{\left[\mathrm{type}\right]}=\mathrm{binary}\\ m_{\mathrm{string}}\left((a^{t},a^{p})\right), & {\left(a^{t},a^{p}\right)}_{\left[\mathrm{type}\right]}=\mathrm{string}\\ m_{\mathrm{decimal}}\left((a^{t},a^{p})\right), & {\left(a^{t},a^{p}\right)}_{\left[\mathrm{type}\right]}=\mathrm{decimal}\\ m_{\mathrm{scale}}\left((a^{t},a^{p})\right), & {\left(a^{t},a^{p}\right)}_{\left[\mathrm{type}\right]}=\mathrm{scale}\\ \mathrm{false}, & \mathrm{otherwise} \end{array}\right.$$
that allows display of matching for any data type and will be used in the following to represent the matching in a single decision node. However, to be able to deal with irrelevance and uncertainty, respective functions for checking these are provided. To also be able to represent these functions in the formal description of the decision making, the Boolean irrelevance check function
(13)$$i\left((a^{t},a^{p})\right)=\left\{\begin{array}{cc} \mathrm{true}, & a_{\left[\mathrm{value}\right]}^{t}\;\mathrm{represents}\;\mathrm{irrelevance}\\ \mathrm{false}, & \mathrm{otherwise} \end{array}\right.$$
is introduced. Additionally, the Boolean MT and particle attribute uncertainty check functions
(14)$$\begin{array}{cc} u_{t}\left(a^{t}\right) & =\left\{\mathrm{true}\;|\;a_{\left[\mathrm{value}\right]}^{t}\;\mathrm{represents}\;\mathrm{uncertainty}\right\} \end{array}$$
(15)$$\begin{array}{cc} u_{p}\left(a^{p}\right) & =\left\{\mathrm{true}\;|\;a_{\left[\mathrm{value}\right]}^{p}\;\mathrm{represents}\;\mathrm{uncertainty}\right\} \end{array}$$
which are encapsulated in the Boolean tuple uncertainty check function
(16)$$u\left((a^{t},a^{p})\right)=\left\{\begin{array}{cc} \mathrm{true}, & u_{t}\left(a^{t}\right)\vee u_{p}\left(a^{p}\right)\\ \mathrm{false}, & \mathrm{otherwise} \end{array}\right.$$
are introduced. A single decision node now can be illustrated as the Boolean decision point function
(17)$$d\left((a^{t},a^{p})\right)=i\left((a^{t},a^{p})\right)\vee u\left((a^{t},a^{p})\right)\vee m\left((a^{t},a^{p})\right)$$
that is modelled via a disjunction of the irrelevance check function i(·), the uncertainty check function u(·), and the major match function m(·) for the tuple $\left(a^{t},a^{p}\right)$. Here, the order of the disjunction represents the relevance of each part. In case of irrelevance no uncertainty check needs to be conducted and in case of uncertainty no matching needs to be conducted. This is due to the need that neither irrelevance nor uncertainty will influence the final decision. The final decision itself for the recommendation of an MT t∈T is determined by the joint decision function
(18)$$D\left(A^{t},A^{p}\right)=\underset{\begin{array}{c} (a^{t},\;a^{p})\in ℜ \end{array}}{⋀}d\left((a^{t},a^{p})\right)=d\left({\left(a^{t},a^{p}\right)}_{1}\right)\wedge \cdots \wedge d\left({\left(a^{t},a^{p}\right)}_{\left|ℜ\right|}\right)$$
that conjuncts the results of any single decision point functions d(·). Given any of the single decision point functions returned true, the joint decision function will also return true. This definition builds the framework on which a decision for the recommendation of an MT t∈T is determined. This framework is encapsulated in the recommendation function
(19)$$r\left(t,p\right)=\left\{\begin{array}{cc} \{t\}, & D(A^{t},A^{p})\\ \emptyset , & \mathrm{otherwise} \end{array}\right.$$
that returns the MT t∈T in case it is recommended for a particle type *p*. Eventually, the joint recommendation function
(20)$$R\left(T,p\right)=\underset{t\in T}{⋃}r\left(t,p\right)=r\left(t_{1},p\right)\cup \cdots \cup r\left(t_{\left|T\right|},p\right)$$
unites any given recommendation on an MT t∈T. The return of R(·,·) represents the set of recommended MTs $T_{R}\subseteq T$ for a given particle type *p*.

Conclusively, the whole mechanism can be summarized as a sequence of parameterized function calls (see Figure 8):
For a given set of MTs *T* and a given single particle type *p* a set $T_{R}$ of recommended MTs is requested via R(T,p).The set of recommended MTs $T_{R}$ is unified based on recommendations of distinct MTs t∈T for the particle type *p* via r(t,p).For any single MT recommendation, the conjunction of all single decision points in *ℜ* is checked via $D(A^{t},A^{p})$.Any single decision point checked in this conjunction is evaluated via $d\left((a^{t},a^{p})\right)$.Specific checks on fulfillment are conducted, represented by a disjunction of an irrelevance check via $i\left((a^{t},a^{p})\right)$, an uncertainty check via $u\left((a^{t},a^{p})\right)$, and a match check via $m\left((a^{t},a^{p})\right)$.

For the special case of a *multi-type* sample consisting out of a set of multiple particle types P⊃{p}, the prior-described decision making process is extended with another step. Here, the results of the several recommendation function calls R(·,·) for any particle type p∈P are intersected by the extended recommendation function
(21)$$\hat{R}\left(T,P\right)=\underset{p\in P}{⋂}R\left(T,p\right)=R\left(T,p_{1}\right)\cap \cdots \cap R\left(T,p_{\left|P\right|}\right)$$
that returns the subset of recommendations $T_{\hat{R}}\subseteq \forall T_{R}$ in which all MTs $t\in T_{\hat{R}}$ are expected to be capable of yielding adequate analysis results for any particle type p∈P of a *multi-type* material. This final step in the decision making sequence might be understood as step 0 (see Figure 8), requiring that recommendations for all particle types p∈P were already inferred.

### 3.4. Analysis Data Import

The e-tool supports the import of a two-column data format for number-based particle size distributions. The first column consists of the upper limit of the particle size classes, the second contains the cumulative distribution function value. This format is featured by the ParticleSizer (https://www.imagej.net/ParticleSizer/ (access 2019-08-16)) [50] (an ImageJ (https://imagej.net/ (access 2019-08-16)) [51] plug-in developed within NanoDefine) for TEM image analysis that can be applied for TEM-based particulate material size measurement approaches [52] and is part of AutoEM [53]. Data generated by the Single Particle Calculation tool (https://www.wur.nl/en/show/Single-Particle-Calculation-tool.htm (access 2019-08-16)) for calculation and evaluation of spICP-MS data [54] is accepted as well. Also, for a powder material the VSSA obtained by the BET method [7] can be given. It should be noticed that a prerequisite for the calculation of the VSSA value is the knowledge of the skeleton density of the powder material. For the case that the skeleton density is unknown, this can be determined by, e.g., Helium pycnometry as a standardized measurement method following ISO 12154:2014 [16]. In addition, manual input of the $x_{50,0}$ is possible.

For these processible inputs an automated nano, non-nano, or borderline decision is determined. Other data will not be processed but may still be attached to the resulting report to support evidence. Supporting a wider variety of analysis data formats was outside the scope of the NanoDefine project; however, it can be established by adding custom implementations of analysis data importers.

### 3.5. Customizability, Extensibility, and Reusability

The NanoDefiner e-tool allows user-individual instrumentation and laboratory settings on MT availability, analysis-related estimated cost and duration, and result measurement uncertainty. Localization and internationalization can be established by altering spreadsheet-based language files. Translations of text blocks into different languages, other than the originally provided localization in British English, can easily be added.

Its knowledge base is entangled with its functionality and buildup. Changes directly affect the decision making and generation of elements and labels in the graphical user interface, for instance those of the MCS. Hence, customization and extension of the knowledge base allow direct changes in the whole e-tool without altering its source code. New material property and MT performance criteria profiles can be added to extend the set of available QCMs, RTMs, and MTs.

Further support of also proprietary analysis data formats can be established by extending the e-tool with custom analysis data importers, using the provided API. Its source code is available on GitHub (https://github.com/NanoDefiner/NanoDefiner (access 2019-08-16)) and was published under the MIT license (https://opensource.org/licenses/MIT (access 2019-08-16)).

## 4. Discussion

Selection of an appropriate MT for the analysis of a specific material requires expert knowledge that is not always available. Hence, the e-tool can support manufacturers of potential nanomaterials during the process of categorization and registration of their products. It is considered in the NANoREG Toolbox [55,56] of the NANoREG project (https://www.nanoreg.eu/ (access 2019-08-16)) as a ready-to-use tool [57] for the implementation of the EC NM recommendation. In addition, beside the NanoDefiner framework [13] and the NanoDefine Methods Manual [42,43,44,45,46], the e-tool is currently recommended by the European Food Safety Authority (EFSA) regarding the European Union food legislation framework [58].

Beside confirmatory Tier 2 MTs, its recommendation of suitable screening Tier 1 MTs provides (under well-defined conditions and within their limits of applicability) alternatives to an analysis with time- and cost-intensive MTs such as TEM that also require their own expertise. Dossier reports generated by the e-tool comprise mandatory and Appendix A required by different registration authorities [39], granting a consolidated representation of the whole process. In addition, the same material categorization is made independent of the operator, e.g., different institutions.

The guided workflow of the e-tool provides information on how to proceed in a walkthrough. The NanoDefine Methods Manual can be accessed anytime for consultation in case further guidance or details on requested input may be necessary. This allows also inexperienced analysts to fulfill a categorization process. Due to the lack of generalizability of SOPs developed during the NanoDefine project [20,46,59,60,61,62], sample preparation knowledge could not be taken into account for MT recommendation. Hence, sample preparation performed in the workflow walkthrough can only be documented but is not considered in the categorization process.

Expert knowledge present in the spreadsheet-based knowledge base can be adjusted and extended by non-computer scientists. For instance, newly added material property and MT performance profiles will be present in the e-tool without altering its source code. It is robust against missing knowledge, allowing it to work efficiently with just the basic set of attributes used for decision making. Manufacturers of materials and MTs may add profiles for their products to test with which MTs their materials can be analyzed adequately, and vice versa, for which materials their MTs are recommended.

The decision making on MT recommendation for the analysis of specific materials is transparent and comprehensible due to explanation on all single decision points. Furthermore, user learning is supported as changes on material properties in the MCS result in live feedback on MT recommendations with explanation. This way, linkage of material properties that lead to exclusion of specific MTs from the recommendation can be revealed. Albeit, as an expert system the e-tool is not intended to substitute human experts but to support them. It is not an adequate surrogate to a human expert possessing creativity and intuition.

Three independent industrial case studies on different and representative industrial materials were conducted to examine whether behavior, recommendations, and decisions yielded by the e-tool are consistent with the NanoDefine DSFS. All case studies approved its concordance [14]. However, results of the rather small number of case studies may not be generalizable. Further case studies on more materials are needed for a generalized statement of consistency.

The e-tool is prepared for anticipated re-definitions of the EC NM recommendation and will be kept up to date concerning this matter. This extends to security aspects, reported bugs, and wording. Hence, its users will not be forced to abandon their installations in the future. On the contrary, an active user community including dialogue, participation, and contributions is the goal.

As it is based on open-source software (see Table A1 in Appendix B) and is open-source itself, the e-tool can be adapted. Its components can serve as templates for other expert systems of the nanoscience domain (but not limited to it). Hence, it may be customized and extended for other purposes. Other projects planning to engineer an expert system may benefit from its reusability. As it was published under the permissive MIT license, its reuse requires no disclosure of derivative work and explicitly allows commercial exploitation.

## 5. Conclusions and Outlook

The NanoDefiner e-tool provides a guided workflow, based on the NanoDefine DSFS [13], that supports analysts with different levels of expertise during the process of material categorization and registration for regulatory purposes: After creating a dossier and describing the particulate material type(s), the user is provided with MT recommendations based on their suitability for the described material. Recommendations are accompanied by detailed and tailored explanation, making decisions transparent and comprehensible. The user then documents applied methods and uploads associated results, an automated categorization is performed for supported analysis file formats. The workflow finishes with the creation of a PDF report aggregating dossier information and measurement results based on a selection of applied methods. Appendix A on MT suitability is appended.

Results of e-tool assessments obtained from independent industrial case studies confirmed its consistency with the underlying NanoDefine DSFS in all cases [14]. In October 2017, the e-tool was released in version 1.0.0 [12]. Currently, it is available in version 1.0.2 and kept up to date in regard to security aspects, reported bugs, wording, and possible revisions of the EC NM recommendation. A publicly accessible service (https://labs.inf.fh-dortmund.de/NanoDefiner/ (access 2019-08-16)) is available for trial purpose. Installation packages and related documentation can be downloaded for local deployment. A virtual machine as well as a Docker (https://www.docker.com/ (access 2019-08-16)) container with installed software packages and default configuration are provided likewise.

Manufacturers of materials and MTs may contribute novel material property and MT performance criteria profiles, providing e-tool users an additional perspective to explore their products. As it is free, open-source, and designed to be extensible and customizable, the e-tool itself or parts of it may serve as template for other knowledge-based expert systems of the nanoscience domain, but not limited to it.

## Figures and Tables

**Figure 1 materials-12-03247-f001:**
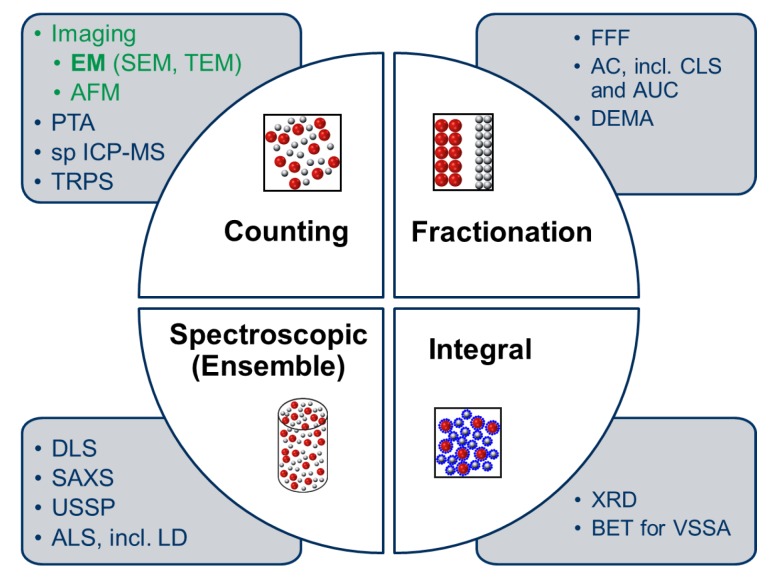
Overview of the measurement techniques (MTs) systematically characterized in the NanoDefine project and included into the NanoDefiner framework. Tier 1 MTs are marked dark blue, Tier 2 MTs are marked green. Displayed acronyms are explained in the text as well as in Table 2.

**Figure 2 materials-12-03247-f002:**
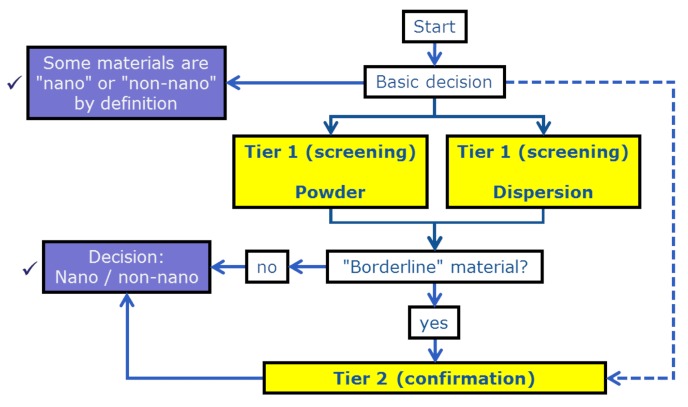
Simplified representation of the NanoDefine Decision Support Flow Scheme. Beside nano and non-nano also a borderline decision is a possible (but intermediate) result of Tier 1 methods. In such cases confirmation via Tier 2 methods is necessary to decide whether a material is nano or non-nano.

**Figure 3 materials-12-03247-f003:**
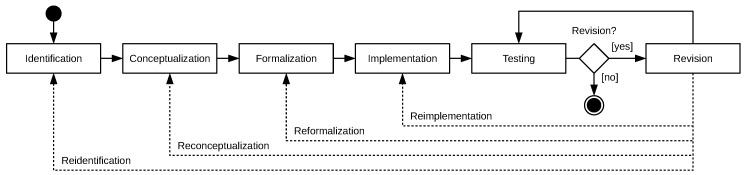
Applied variant of a classical knowledge engineering process, adapted from [32].

**Figure 4 materials-12-03247-f004:**
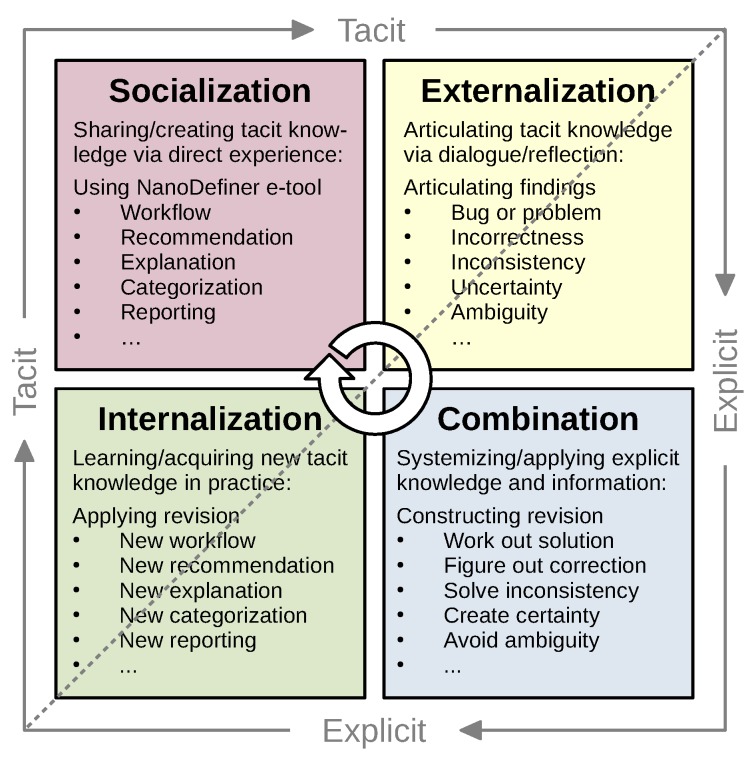
Knowledge management during the development of the NanoDefiner e-tool. Revision cycles were oriented towards the *Socialization* (purple), *Externalization* (yellow), *Combination* (blue), *Internalization* (green) model of knowledge creation, adapted from [34].

**Figure 5 materials-12-03247-f005:**
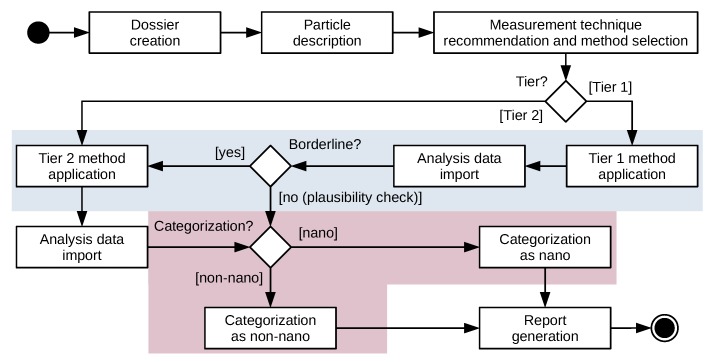
Simplified representation of the guided workflow in the NanoDefiner e-tool, highlighting Tier 1 to Tier 2 escalation in case of a borderline material (blue), and the final categorization step (purple).

**Figure 6 materials-12-03247-f006:**
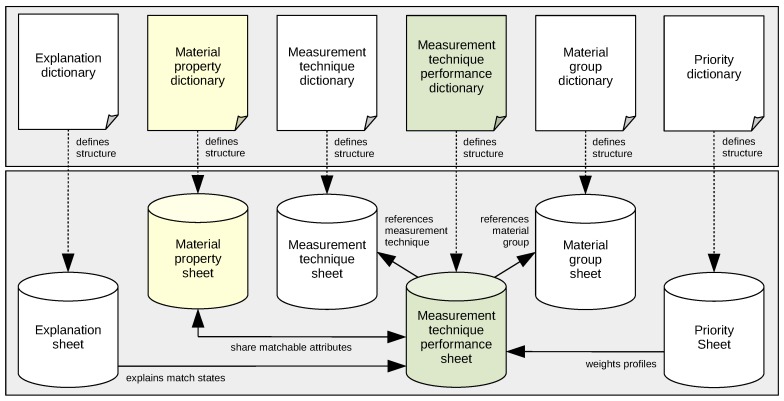
Knowledge base setup with data dictionaries (upper part) and their associated data sheets (lower part), showing core components (yellow, green) and administrative components (plain).

**Figure 7 materials-12-03247-f007:**
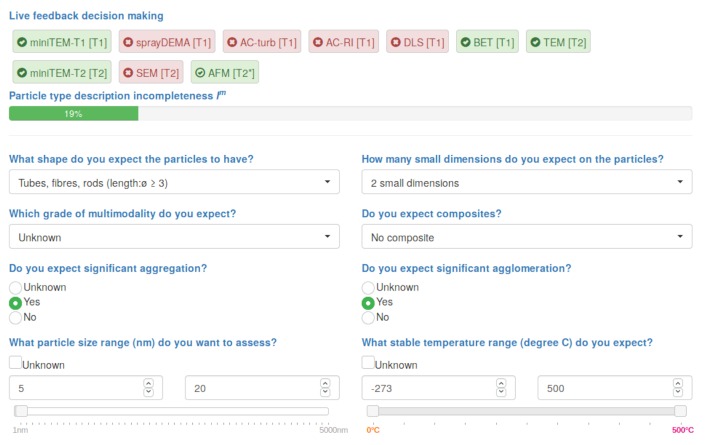
An excerpt of the material categorization scheme (MCS) via which physicochemical particle type properties are described (lower part). Changes in the MCS trigger live feedback on recommendations (upper part): Recommended measurement technique are highlighted green with a check mark, unrecommended are highlighted red with a cross mark.

**Figure 8 materials-12-03247-f008:**
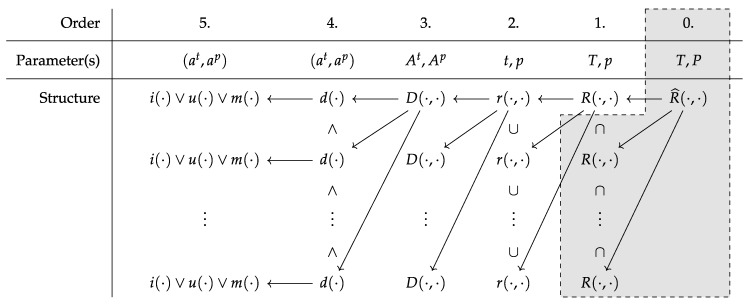
Illustration of function calls with parameterization of the formally described decision making process for a single particle type (excluding the gray area), respectively for multiple particle types (including the gray area).

**Table 1 materials-12-03247-t001:** Quality control and representative test materials considered in the NanoDefine project.

Quality Control or Representative Test Material	Size Range	NanoDefine Code	Alternative Code [6]
Quality Control Materials			
Colloidal Au (monomodal)	15–30 nm	ID-16	QCM3
Colloidal SiO${}_{2}$ (monomodal)	approx. 15–50 nm	ID-17	QCM2
Colloidal SiO${}_{2}$ (trimodal)	approx. 20–200 nm	ID-18	QCM6
Polystyrene (monomodal)	approx. 30–70 nm	ID-19	QCM1
Polystyrene (trimodal)	approx. 30–400 nm	ID-20	QCM5
Colloidal Ag (monomodal)	3–10 nm	ID-21	QCM4
Representative Test Materials			
Zeolite	4–900 nm ${}^{a}$	BAM-11	–
Organic pigment Y83 (nano grade)	approx. 20–100 nm	IRMM-380	RTM7
BaSO${}_{4}$ (fine grade)	approx. 0.1–2 µm	IRMM-381	RTM2
Multi-walled carbon nanotubes	approx. 5–25 nm ${}^{b}$	IRMM-382	–
Nano steel	approx. 0.05–1.5 µm	IRMM-383	–
CaCO${}_{3}$ (fine grade)	approx. 50–500 nm	IRMM-384	RTM4
Kaolin	approx. 0.04–2 µm ${}^{c}$	IRMM-385	RTM5
Organic pigment Y83 (coarse grade)	approx. 50–800 nm	IRMM-386	RTM8
BaSO${}_{4}$ (ultrafine grade)	approx. 10–150 nm	IRMM-387	RTM1
TiO${}_{2}$ (coated) ${}^{d}$	approx. 60–600 nm	IRMM-388	RTM3
Basic methacrylate copolymer	approx. 0.3–10 µm ^e^	IRMM-389	RTM9

${}^{a}$ It is not clear whether constituents or aggregates were measured; ${}^{b}$ Estimate refers to thickness of multi-walled carbon nanotubes; ${}^{c}$ Estimate of the kaolin platelets is poor (partly thickness, partly front diameter); ${}^{d}$ Coating consists of silicon and aluminum compounds/oxides and is not further specified by the manufacturer (thin, inorganic layer of 3–5 nm, strongly adhering to TiO${}_{2}$ (rutile) core); ^e^ It is not clear what was measured, contradictory results due to difficulties in sample preparation.

**Table 2 materials-12-03247-t002:** Measurement techniques considered in the NanoDefine project, tiered into screening (Tier 1) and confirmatory (Tier 2) techniques.

Measurement Technique Name	Short Name	Tier	Assessed
Tier 1: Screening			
Analytical Centrifugation, turbidity and refractive index	AC-turb/-RI	1	Yes
Angular Light Scattering	ALS	1	No
Asymmetric Flow FFF, Multi-angular Light Scattering	AF4-MALS	1	No
Brunauer–Emmett–Teller	BET	1	Yes
Dynamic Light Scattering	DLS	1	Yes
Mini Transmission Electron Microscopy with additional Tier 2 mode	miniTEM	1	Yes
Particle Tracking Analysis	PTA	1	No
Single Particle Inductively Coupled Plasma Mass Spectrometry	spICP-MS	1	No
Small-Angle X-ray Scattering	SAXS	1	No
Spray Differential Electrical Mobility Analysis	sprayDEMA	1	Yes
Tunable Resistive Pulse Sensing	TRPS	1	No
Ultrasonic Spectroscopy	USSP	1	No
X-ray Diffraction	XRD	1	No
Tier 2: Confirmatory			
Atomic Force Microscopy	AFM	2	No
Scanning Electron Microscopy	SEM	2	Yes
Transmission Electron Microscopy	TEM	2	Yes

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
