# Peer review of "NanoDefiner e-Tool: An Implemented Decision Support Framework for Nanomaterial Identification"

_materials, 2019, doi:10.3390/ma12193247_

Round 1

Reviewer 1 Report

The article describing NanoDefiner e-Tool is giving all the details of this tool. Every step is presented and it could be a useful tool for those working with nanoparticles.

Small corrections:

Ø  Line 32 – maybe it might be useful for readers to provide a definition of the term “number-weighted size distribution (x50,0)”

Ø  Lines 138-141: the phrase ”The nano/non-nano cutoff value of the particle number-weighted median as resulted from investigations in the NanoDefine project, set at 250 nm to escalate from Tier 1 to Tier 2, can be adapted at a later stage within the e-tool as a possible consequence of upcoming, improved systematic studies” it's a bit long. May it could be reformulated, for example: “The nano/non-nano cutoff value of the particle number-weighted median (as resulted from investigations in the NanoDefine project) is set at 250 nm to escalate from Tier 1 to Tier 2. This value can be adapted at a later stage within the e-tool as a possible consequence of upcoming, improved systematic studies.”

Ø  Lines 173-176: the phrase “Explicit concepts on the e-tool regarding its workflow, decision making on MT recommendations for the analysis of specific materials, and categorization of potential nanomaterials were created by knowledge engineers, considering the main sources of knowledge and requirements identified priorly” it's a bit long. May it could be reformulated.

Ø  Line 340: “Table S4, and Table S4 in Supplementary Materials” maybe could be replaced by “Table S3, and Table S4 in Supplementary Materials”

Reviewer 2 Report

The manuscript reports on an open source software developed within a European project NanoDefine which aims to develop tools for categorizing materials as nano or non-nanomaterials. Software called NanoDefiner e-tool was developed for users with minimal experience in material characterization and computer technology to be able to categorize materials or identify suitable particle size characterization methods.

The topic is well suited for the journal and the described e-tool, although still in the test phase, is already considered for application in a NANoREG project and European Food Safety Authority which speaks for the tool’s usefulness in implementation of the EC nanomaterial recommendation.

The manuscript is well written and only needs minor improvements in language and presentation.

Keywords: “European Commission’s recommendation on the definition of nanomaterial” is a bit long for one keyword, could be broken down to several keywords.

Introduction

The first sentence does not sound grammatically correct. “EC’s recommendation… has made systematic investigations”? Please rephrase.

One aspect that could be outlined in the beginning of the introduction, is the EC’s definition of “nanomaterial”, so it would be clear that the definition is based on particle size. It would also be important to explicitly state that NanoDefine focuses on the size measurements of particulate materials for categorization purposes. This would eliminate any confusion or misunderstanding about the parameter(s) what is (are) being used to define nanomaterials and non-nano materials. For example, there have been suggestions to include other material characteristics, such as bioactivity, for categorization of materials.

The sentence: “Results were cross-checked against values obtained by electron microscopy as the reference method.” (lines 29-30), is unclear in its current location. It is unclear which results are being referred to. This sentence may be better moved after the next sentence. The beginning of the introduction would benefit from more explicit explanation of the project aims, scope and approach.

Line 34, The sentence starting with “Another major project outcome…” is a bit confusing, because it is not clear from the writing in the first paragraph what the first project outcome was.

In line 56, and also in the conclusions the authors mention independent industrial case studies [13] which seem not to be available to the reader but would be necessary to illustrate the applicability of the e-tool. Is there a plan to publish the case studies separately? If not, it would be necessary to include these in the current paper as supporting material.

Materials and Methods

Materials and Methods part seems to be written in a discussion style, meaning that it contains excessive text that doesn’t serve the purpose of clearly stating how something was done. For example, the first paragraph (lines 60-66) is redundant and should be deleted or moved to introduction or discussion instead.

The text in lines 86-90 does not belong to materials and methods.

Table 1. Please add explanations what “monomodal” and “trimodal” mean in the context of this paper. For representative test materials, please add size ranges for the “nano”, “fine”, “coarse” and “ultrafine” grades and explain what “coated” means at TiO2.

Figure 1. Add a note to the caption that acronyms are explained in the text and Table 2.

Lines 101-102, please rephrase or explain “more versions” at centrifugation techniques.

Line 106, “with knowledge of skeleton density” is unclear.

Line 128, replace “(see Figure 2) can be summarized as followed.“ with “is depicted in Figure 2.” or similar.

Results

Lines 213-214, replace “addresses” with “accommodates” or “suites” or similar, and replace “groups” with “categorizes” or “divides” or similar.

Line 279, Correct “For description the MTs,” (For description of the MTs?)

Discussion

Line 447, replace “not available anywhere, anytime” with “not always available.”

Lines 460-461, the statement that also inexperienced analysts would be able to fulfill the categorization process raises a question if the parameters that need to be entered in the e-tool are explained and defined at the respective input slots. This should be stated in the manuscript. For example, two parameters "aggregation" and "agglomeration" are often used interchangeably in the literature, so these terms need to be well explained and defined. Inexperienced users may not be able to differentiate the two characteristics.
